# Epidemiology of Mucormycosis in Greece; Results from a Nationwide Prospective Survey and Published Case Reports

**DOI:** 10.3390/jof9040425

**Published:** 2023-03-29

**Authors:** Maria Drogari-Apiranthitou, Anna Skiada, Ioannis Panayiotides, Timoleon-Achilleas Vyzantiadis, Aikaterina Poulopoulou, Myrto Christofidou, Anastasia Antoniadou, Emmanuel Roilides, Elias Iosifidis, Vassiliki Mamali, Athina Argyropoulou, Styliani Sympardi, Nikoletta Charalampaki, Nikolaos Antonakos, Paraskevi Mantzana, Zafeiria Mastora, Ourania Nicolatou-Galitis, Maria Orfanidou, Zoi-Dorothea Pana, Ioannis Pavleas, Angelos Pefanis, Vissaria Sakka, Anastasia Spiliopoulou, Maria Stamouli, Polydoros Tofas, Eleni Vagiakou, George Petrikkos

**Affiliations:** 1Infectious Diseases Research Laboratory, 4th Department of Internal Medicine, Attikon General University Hospital, Medical School, National and Kapodistrian University of Athens, 12462 Athens, Greece; 21st Department of Internal Medicine, National and Kapodistrian University of Athens, Laiko General Hospital, 11527 Athens, Greece; 32nd Department of Pathology, National and Kapodistrian University of Athens, Medical School, Attikon University Hospital, 12462 Athens, Greece; 4Department of Microbiology, Medical School, Aristotle University of Thessaloniki, 54124 Thessaloniki, Greece; 5Department of Microbiology, University Hospital of Patras, 26504 Patras, Greece; 6Infectious Diseases Unit, 3nd Department of Paediatrics, School of Medicine, Aristotle University and Hippokration General Hospital, 54642 Thessaloniki, Greece; 7Department of Microbiology, Tzaneio General Hospital, 18536 Piraeus, Greece; 8Department of Clinical Microbiology, Evangelismos General Hospital, 10676 Athens, Greece; 91st Department of Internal Medicine, Thriasio General Hospital of Eleusis, 19600 Eleusis, Greece; 10Clinical Microbiology Laboratory, Thriasio General Hospital of Eleusis, 19600 Eleusis, Greece; 11Department of Microbiology, AHEPA University Hospital, 54636 Thessaloniki, Greece; 121st Department of Critical Care Medicine and Pulmonary Services, Evangelismos Hospital, National and Kapodistrian University of Athens Medical School, 10676 Athens, Greece; 13CureCancer, 15451 Athens, Greece; 14Clinical Microbiology Laboratory, General Hospital of Athens Georgios Gennimatas, 11527 Athens, Greece; 15School of Medicine, European University of Cyprus, Nicosia 2404, Cyprus; 16Intensive Care Unit, Laiko General Hospital, 11527 Athens, Greece; 17Department of Internal Medicine, Sotiria General and Chest Diseases Hospital of Athens, 11527 Athens, Greece; 183rd Department of Internal Medicine, Sotiria General and Chest Diseases Hospital of Athens, 11527 Athens, Greece; 192nd Department of Internal Medicine, Propaedeutic, Haematology Unit, Attikon University General Hospital, Medical School, National and Kapodistrian University of Athens, 12462 Athens, Greece; 20Athens Euroclinic, 11521 Athens, Greece

**Keywords:** mucormycosis, zygomycosis, Mucorales, epidemiology, Greece

## Abstract

Mucormycosis has emerged as a group of severe infections mainly in immunocompromised patients. We analysed the epidemiology of mucormycosis in Greece in a multicentre, nationwide prospective survey of patients of all ages, during 2005–2022. A total of 108 cases were recorded. The annual incidence declined after 2009 and appeared stable thereafter, at 0.54 cases/million population. The most common forms were rhinocerebral (51.8%), cutaneous (32.4%), and pulmonary (11.1%). Main underlying conditions were haematologic malignancy/neutropenia (29.9%), haematopoietic stem cell transplantation (4.7%), diabetes mellitus (DM) (15.9%), other immunodeficiencies (23.4%), while 22.4% of cases involved immunocompetent individuals with cutaneous/soft-tissue infections after motor vehicle accident, surgical/iatrogenic trauma, burns, and injuries associated with natural disasters. Additionally, DM or steroid-induced DM was reported as a comorbidity in 21.5% of cases with various main conditions. *Rhizopus* (mostly *R. arrhizus*) predominated (67.1%), followed by *Lichtheimia* (8.5%) and *Mucor* (6.1%). Antifungal treatment consisted mainly of liposomal amphotericin B (86.3%), median dose 7 mg/kg/day, range 3–10 mg/kg/day, with or without posaconazole. Crude mortality was 62.8% during 2005–2008 but decreased significantly after 2009, at 34.9% (*p* = 0.02), with four times fewer haematological cases, fewer iatrogenic infections, and fewer cases with advanced rhinocerebral form. The increased DM prevalence should alert clinicians for timely diagnosis of mucormycosis in this patient population.

## 1. Introduction

Mucormycosis has emerged as an important group of fungal infections in the past two decades, affecting mainly immunocompromised patients. It is caused by mucormycetes (previously zygomycetes), fungi belonging to the order Mucorales. Mucormycosis progresses rapidly, leading to fungal angioinvasion with subsequent thrombosis and tissue necrosis. Major risk factors are haematologic malignancies, organ or bone marrow transplantation, neutropenia, poorly controlled diabetes mellitus (DM), treatment with corticosteroids, and deferoxamine therapy [1]. Infection with SARS-CoV-2 has emerged as a new risk factor for the development of mucormycosis, as evidenced mainly in India during the second wave of the COVID-19 pandemic [2]. Trauma or burns are major risk factors for immunocompetent patients [3,4]. Mucormycosis causes high morbidity and mortality unless it is diagnosed early and treated appropriately. Its prevalence and incidence vary from country to country and are not easy to calculate because it is a relatively rare disease and population-based data are scarce. 

The epidemiology of mucormycosis in Greece is not known so far. In previous years, only case reports were published. In 2003, in a retrospective study, Petrikkos et al. acknowledged the increasing frequency of these infections in Greece [5]. In 2005, a nationwide multicentre prospective survey on mucormycosis was initiated, and is still ongoing. The present study aims to present and analyse the data collected from January 2005 until the end of 2022. For a better estimate of the burden of the disease, data from published case reports that occurred before 2005, or after 2005 but were not recorded in the survey, are also presented.

## 2. Materials and Methods

### 2.1. Case Lists

#### 2.1.1. Prospective Study

Upon suspicion or definitive diagnosis of a case there was a voluntary contact with one of the coordinators of the study, either at the Infectious Diseases Research laboratory/4th Department of Medicine, Attikon General University Hospital, or at the 1st Department of Medicine, Laiko General Hospital, National University of Athens. Each case was recorded in a case report form which was sent to our Laboratory or was directly submitted to the zygomyco.net (official site of the ECMM/ISHAM Working Group on Zygomycosis) database [6] and retrieved by one of the coordinators in real-time. The isolated fungi were also sent to our laboratory if the participating hospitals did not have the means for identification to genus or species level. For cases with negative cultures or when these were not performed, formalin fixed, and paraffin embedded (FFPE) tissue blocks were used for molecular confirmation of the diagnosis. Eleven tertiary care academic and state hospitals participated in the present study, with cases from almost all regions of Greece.

Participants followed instructions for proper specimen storage and culture conditions required for optimal growth of mucormycetes [7]. 

The collected data included demographic and clinical characteristics of the patients, underlying conditions and risk factors, time and type of laboratory diagnosis, treatment, and outcome. Only proven and probable infections according to the revised EORTC/MSG criteria [8] were registered. 

#### 2.1.2. Published Case Reports

Mycormycosis is not a notifiable disease, therefore, for a better overview of the burden of these infections, a literature review was also performed, searching the PubMed^®^ and Scopus^®^ databases, as well as the Greek database Iatrotek which includes all articles published in the main Greek medical journals since 1981. The terms used were “mucormycosis AND Greece”, zygomycosis AND Greece”, “mucor AND Greece”, “Rhizopus AND Greece”, “Absidia AND Greece”, and “mucor”, “Absidia”, “Rhizopus”, “φυκομύκωση”, “μουκορμύκωση”, “μουκορμύκωσις” for the international and Greek data bases, respectively. 

The data from the retrieved published reports were used together with the data from the prospective survey only for the calculation of the annual incidence. For all other parameters, data were analysed separately, in order to avoid publication bias.

The two case lists were compared for duplicates. Published cases that were already registered [9,10,11,12,13,14], were removed from the published case list. 

### 2.2. Incidence

The annual incidence was calculated taking into consideration the sum of the cases reported in the prospective registry, and those retrieved from the literature since 2005, when the survey was initiated. The general population was used as denominator, based on the 2011 population-housing census (~11 million) [15].

### 2.3. Laboratory Diagnosis

Laboratory diagnosis was based on histopathological and microbiological methods, including direct microscopy, culture, and molecular methods. Histopathological stains included both standard Haematoxylin and Eosin (H&E) and special stains (Periodic Acid—Schiff [PAS] and silver impregnation stains, e.g., Grocott). For the identification of mucormycetes from formalin-fixed and paraffin-embedded tissues (FFPE), a semi-nested PCR method was applied [16,17]. 

Direct microscopy of wet tissue mounts was performed with potassium hydroxide (KOH 10–20%) with or without an optical brightener (Blankophor^®^ 0.025% *w*/*v*).

Tissue samples or other materials were cultured on media and under conditions as per standard of care, including Sabouraud dextrose agar with chloramphenicol, without cycloheximide, at 30 °C and 37 °C. Grinding of specimens was avoided [7].

Identification was based on standard mycological methods such as macroscopic and microscopic morphology of the colonies and culture techniques. Molecular analysis for confirmation was performed in most cases, and was carried out partly at the Mycology Reference Centre, National Centre for Microbiology, Madrid, Spain, in the framework of a collaboration with the European Confederation of Medical Mycology (ECMM) working group for the registry of zygomycosis in Europe, and partly at the Westerdijk Fungal Biodiversity Institute, Utrecht, the Netherlands (formerly Centraalbureau voor Schimmelcultures, CBS), and isolates were deposited at its collection. Primers V9G and LR3 were used to amplify the complete ITS region and the D1/D2 region of the LSU, as described [18]. For sequencing, the primer set ITS1 and ITS4 for the complete ITS region, and NL1 and LR3 for the D1/D2 region of the LSU were used [18]. Most of cases from Northern Greece were molecularly confirmed at the Department of Microbiology, Medical School, Aristotle University of Thessaloniki, Greece, in a panfungal PCR for the amplification of the ITS 1 or, less frequently the ITS 2 region, followed by standard Sanger sequencing. To compare single ITS sequences the ISHAM barcoding database (https://its.mycologylab.org/, accessed on 30 January 2023) was used [19].

### 2.4. Statistical Analysis

The two-tailed Fisher’s exact test was used for the comparison of categorical variables. *p* values < 0.05 were considered significant.

## 3. Results

### 3.1. Demographic Data, Incidence

From January 2005 to December 2022, 108 cases were reported, of which 60 were submitted to the zygomyco.net database [6]. 

Eight cases (7.4%) involved children. Of the 100 adults, 61 were men (male-to-female ratio 1.3:1), and the median age was 61 years (range 18–87 years). The children were five boys and three girls with a median age of 3 years (range 1–16 years). 

From the literature search, another 74 cases were retrieved (Table 1) [20,21,22,23,24,25,26,27,28,29,30,31,32,33,34,35,36,37,38,39,40,41,42,43,44,45,46,47,48,49,50,51,52,53,54,55]. Adults were 63, and 11 were children (14.9%). Of the adults, 39 were men (male-to-female ratio 1.1:1) and median age was 57 years (range 22–87 years).

The annual incidence was estimated to be 10.7 cases/year on average (0.97 cases/million population) for the whole study period. As seen in Figure 1, a decline in annual incidence was observed after 2009, at an average of 6 cases/year (0.54 cases/million population). 

### 3.2. Prospective Study

#### 3.2.1. Localisation/Clinical Forms

Most cases had the “rhinocerebral” clinical form, the term including infections localised only on the sinus or palate, and those with rhino-orbital, rhino-cerebral, rhino-orbital-cerebral, or only orbital involvement (56 cases, 51.8%) (Table 2).

Because of the noted decline of the incidence during the study, the data between the periods 2005–2008 and 2009–2022 were compared (Table 2) and no substantial differences in the clinical forms were detected.

#### 3.2.2. Underlying Conditions

Detailed results are shown in Table 3. The percentages were calculated on 107 patients, as one patient had two infections with different organisms three years apart, but the same underlying condition (nasal polyps). The leading underlying condition or risk factor for mucormycosis was haematologic malignancy/neutropenia (32 cases, 29.9%), mostly acute myeloid leukaemia (AML). Another five patients (4.7%) had undergone allogeneic haematopoietic stem cell transplantation (HSCT). Two cases had conditions leading to haemochromatosis: an infant with Pearson syndrome, in whom mucormycosis developed around an intravenous catheter insertion site [9], and a patient with transfusion-dependent β-thalassemia (TDT), both with iron overload and treatment with deferoxamine. 

Diabetes mellitus (DM) type 2 was the main underlying condition in 17 (15.9%) cases. In addition, it was reported as comorbidity in other 11 patients (10.3%) with various main conditions. Furthermore, at least 30 (28%) patients were previously treated with corticosteroids, among them 19 with haematologic malignancy. Twelve patients developed steroid-induced diabetes (11.2%). 

The patients with trauma as a predisposing factor were 24 (22.4%) and all had the cutaneous form. Of these, 22 were immunocompetent and two had DM. In 15 out of 24, mucormycosis was iatrogenic, secondary to surgery such as spinal fixation, grafting ascending aorta, vascular surgery, knee arthroplasty, surgical wound, or after central venous catheter insertion or vein puncture (six patients). Eight of the above infections occurred in patients who had undergone vascular or other surgery in the same hospital during the years 2006–2007 and were related to contaminated patches covering the surgery wounds. Two further patients sustained burn injuries, one of them during the wildfire in Mati, Attica, in July 2018. Another patient suffered injuries during a flash flood after heavy rain in Mandra, Attica, in November 2017. Six patients sustained injuries mostly due to motor vehicle accidents, one of them agricultural.

Autoimmune disorders included systemic lupus erythematosus (SLE), vasculitis with granulomatosis (Wegener disease), psoriatic arthritis, IgA nephropathy, and immune thrombocytopenic purpura (ITP) (eight patients, 7.5%). All of these patients had received corticosteroids and two of them had neutropenia.

Chronic renal failure and haemodialysis was the main underlying condition in one patient and was the comorbidity of the main underlying disease in three. 

The clinical forms in haematological and diabetic patients are shown in Table 4. The rhinocerebral form was the most prevalent in both of these conditions. 

When the periods before and after 2009 were compared, proportionally fewer haematological and fewer trauma cases were observed during the second period (Table 3), whereas the proportions of cases with diabetes and solid organ (SO) cancer were increased. Looking at the patient absolute numbers, however, we noticed that cases of all underlying conditions were decreased annually, though each at a different rate. More specifically, there were five times fewer trauma cases, four times fewer haematological cases, and about half DM and SO cancer cases annually during 2009–2022, compared to the period 2005–2008. In the second period, only two iatrogenic cutaneous cases were reported. 

Further differences regarding the qualitative characteristics of the infections between the two periods are depicted in Figure 2. The two main underlying conditions, haematologic malignancies and DM, are shown. In the second period, the haematological patients presented more often with less advanced infections of the rhinocerebral form, usually localised in the sinuses or palate without brain involvement. The patients with DM had less cutaneous but more sinus infections during the second period. 

#### 3.2.3. Laboratory Diagnosis

Diagnosis was confirmed by histopathological examination of tissue specimens in 85 cases (78.7%). Direct microscopy of wet tissue mounts was performed in 50 cases (46.3%), all but one found positive. Culture was performed in 95 (88%) cases. 

#### 3.2.4. Infectious Agents

Of the 95 cultured cases, 81 (85.3%) were positive. In 14 (14.7%), no fungal growth was observed. In one case with negative culture, the fungus was identified molecularly from the FFPE tissue. The most frequent genus was *Rhizopus* (55 cases, 67.1%). *R. arrhizus* (syn. *R. oryzae*) was responsible for most infections (45, 54.9%). Other genera/species isolated were *Lichtheimia* (formerly *Absidia corymbifera*), *Mucor*, *Saksenaea vasiformis*, *Apophysomyces elegans*, and *Syncephalastrum racemosum* (Table 5). All other unidentified mucormycetes are designated as “Mucorales”. There were less unidentified Mucorales during the second period (eight and two isolates in the periods before and after 2009, respectively).

Site of isolation for each fungal genus and all-cause mortality according to genus are also presented in Table 5. *R. arrhizus* was isolated from 36 out of 43 (83.7%) cases of rhinocerebral mucormycosis with a positive culture, whereas *Lichtheimia* and *Mucor* sp. from only one case each. In the remaining five cases, the isolated Mucorales were not further identified. In five cases, another fungus was simultaneously isolated, but this was histologically proven only in one case, a road accident trauma infected with *S*. *racemosum* and *F*. *solani* [14].

#### 3.2.5. Therapy

All patients were treated unless the diagnosis was made at an end-stage of their disease or post-mortem (three cases). The antifungal therapy was reported in 73 cases. Thirty-one (42.5%) patients were treated with Liposomal amphotericin B (L-AmB, AmBisome^®,^, Gilead Sciences Hellas, Athens, Greece) alone (median dose 7 mg/kg/day, range 3–10 mg/kg/day) and 32 (43.8%) received also oral posaconazole (400 mg × 2/day), in most cases (21/32, 65.6%) after completion of the treatment with L-AmB and for maintenance after hospital discharge. Five patients received L-AmB combined with another antifungal (micafungin, anidulafungin, caspofungin, voriconazole) usually due to previous or simultaneous infections with other fungal agents such as *Candida*, *Aspergillus*, or *Fusarium*, and one received L-AmB with isavuconazole. Three received amphotericin B lipid complex (Abelcet^®^), and one was treated with posaconazole alone. 

The median duration of antifungal therapy in the cured was 38 for L-AmB (range 13 days to 5 months) and 60 days for posaconazole (range 9 days to 11 months).

Surgery was performed in 53 (49.1%) patients, usually requiring repeated debridements. In 22 (20.4%) patients, surgery was not performed for the following reasons: (i) the infection was too extensive, and an operation would be incompatible with life, (ii) the underlying condition would not permit it (e.g., thrombocytopenia), (iii) the patient was at the end-stage of disease. Only two of the patients without surgery survived. There was no relevant information for the remaining patients. 

#### 3.2.6. Outcome

The outcome at 60 days was known for 86 patients. Of these, 44 (51.2%) survived, or were cured or improved, and 42 died (48.8% all-cause mortality). Death cases per year are shown in Figure 3. During the years 2005–2008, the fatalities were 27 out of 43 cases with known outcome (62.8%). During 2009–2022, the cases with known outcome were also 43, of which 15 were fatal (34.9%, *p* = 0.02). As seen in Table 6, death was highly dependent upon the site and the extension of infection. In cases with disseminated infection, the mortality was 100%, followed by the rhino-(orbital)-cerebral form (91% before 2009, 73% after 2009). Among the patients with sinusitis without orbital or brain involvement, only one of the 14 with known outcome died after 2009 (28.6%, vs. 7.1% before, and after 2009, respectively). However, this difference was not statistically significant (*p* = 0.25). The same was true for all categories shown in Table 6, but it should also be noted that the sample size per category was very small. 

The time from diagnosis to death was usually short, with a median of 11 days, ranging from one to 60 days. Seven more patients died shortly thereafter, usually at six months, due to the underlying disease or cardiac arrest. 

### 3.3. Published Case Reports

Detailed data can be seen in Table 1. Summarised data, as well as differences between published cases after 2005 (*n* = 32) and registered cases, are shown in Table 7.

Of the trauma cases, four were iatrogenic, constituting an outbreak in a haematology unit. These were associated with adhesive polyethylene tapes used to stabilise peripheral venous catheters in the patients’ forearms [45].

Histology was performed in 66 (89.2%) of the published cases and was not reported in the remaining eight.

Surgery was performed in 41 cases (55.4%). A remarkable case was that of a 16-year-old female kidney transplant recipient with disseminated mucormycosis, who survived [51]. The patient received 1 month of intravenous liposomal amphotericin B followed by 1 month of oral posaconazole, which resulted in an excellent clinical response, while renal function remained stable.

## 4. Discussion

Timely diagnosis of mucormycosis is crucial for survival, but it requires a high index of suspicion, which is difficult for rare infections. Knowledge of the population-at-risk and the local epidemiology helps to identify the disease early, so that prompt and proper therapy can be initiated. Registries are useful tools to that end. A survey for the cases of mucormycosis in Greece was initiated in 2005, soon after the ECMM Working Group on Zygomycosis and the European registry initiative were formed (now global registry of the ECMM/ISHAM Working Group on Zygomycosis [6]). 

The first case-report of mucormycosis in Greece was published in 1984 [20], on a patient with DM. Until 2005, another 40 cases were published, including the patients in the study by Petrikkos et al. [5,20,21,22,23,24,25,26,27,28,29,30,31,32,33,34,35,36,37,38,39,40,41,42,43,44,45,46,47,48,49,50,51,52,53,54,55], most of them after 2002. Based on the present data, both from registered and published cases after 2005, the overall average incidence rate per year was calculated to be 0.97/million population. However, except for a substantial increase during the years 2006 and 2007, the incidence decreased and remained more or less stable thereafter, at 0.54/million/year, or six patients/year. The incidence of mucormycosis in most countries is unknown. The few existing population-based studies [56,57,58,59] differ in capture periods, populations, and definition or diagnostic procedures. However, they all show an increasing incidence in the past two decades. In Europe, in a retrospective study in France, an increase from 0.7/million in 1997 to 1.2/million in 2006 was found [56]. In Spain, a similar increase was found in a retrospective longitudinal descriptive study for the period 1997 to 2018 [59], and in Belgium, the incidence of mucormycosis in a large university hospital also increased during 2000–2009 [60]. Rough estimates from other European countries show a lower annual incidence, ranging from 0.2 to 0.9 cases/million population [61,62,63], similar to the present study. In the Asian continent on the other hand, the incidence is much higher, with a prevalence in India estimated to be about 70 times higher than globally [64]. 

Haematologic malignancy (HM) was the most frequent underlying condition in the present survey compared to DM as the main underlying disorder. This is in line with data from other European countries, Turkey, and Australia [65,66,67,68,69]. DM, however, also complicated cases with other main underlying diseases in our cohort, and some additional cases had corticosteroid-induced DM. Taking all these data together, a percentage of 37.4% of cases with DM is reached, highlighting the importance of DM in the development of mucormycosis. In studies from the Middle East and North Africa, Iran, India, Mexico, and the USA, DM was the most common underlying condition [58,70,71,72], with percentages close to 75% in Iran and India [70,71]. 

A high percentage of patients with DM and mucormycosis may also reflect the prevalence of DM in a population. Based on the results from the recent national Health Examination Survey (HES) on the prevalence of diabetes in Greece, the overall DM prevalence was 11.9%, including patients known to have diabetes (10.4%), and patients diagnosed during the study (unknown diabetes, 1.5%), and has apparently increased in recent years [73]. Of those with known diabetes (and measured HbA1c), 70.9% were well controlled (HbA1c < 7.0) [73]. In other European countries such as France, Italy, Germany, and Spain, DM prevalence was found to be 6%, 7%, 8.9%, and 13.8%, respectively [74,75,76,77]. In India, the country with the second largest diabetic population globally after China, DM prevalence varied from 9.3% to 16.6%, according to geographic area [78], but more importantly, the levels of awareness, treatment, and control, were very low [79]. India has a long history of high prevalence of mucormycosis. Furthermore, the unprecedented increase in cases of Coronavirus-disease associated mucormycosis (CAM) during the second wave of the COVID-19 pandemic in India, much higher than in the rest of the world [2], was strongly associated with the presence of diabetes and use of systemic steroids [80]. In fact, by the end of 2021, more than 45,000 CAM cases were reported in India, and very few in other 17 countries, mostly single cases [2]. No CAM cases have been reported in Greece so far.

Comparing the periods before and after 2009, four times less haematological cases with mucormycosis were observed during the second period. The use of posaconazole, an azole with significant anti-Mucorales activity, for invasive fungal infection prophylaxis in eligible patients with AML or myelodysplastic syndrome (MDS) [81,82,83,84,85] at this time, may have contributed to the decrease. 

Children represented only a small proportion of the cases. The main risk factors in this age group were haematologic malignancy, SO cancer, and kidney transplantation, in line with collective and unique data from the Zygomyco.net [6] and FungiScope™ (University of Cologne, Cologne, Germany) registries [86].

Most cases with HM or DM had the rhinocerebral form in our study. This is a remarkable result, as previous data had shown that the pulmonary form of mucormycosis was predominant in haematological patients and the rhinocerebral in patients with DM [66]. Analysis of our cases according to the progression of this form, from localisation on the sinus or palate only, to a large extension affecting the orbit and brain, showed fewer cases with a large extension during the second period of the study. The localised infections were treatable with surgery and antifungals, therefore they had also significantly less mortality. Increased awareness, both in the clinic and the laboratory in recent years, might have contributed to improved outcomes.

Pulmonary is the second most common form of mucormycosis globally, often seen in patients with haematologic malignancies and in transplant recipients [87,88]. Only a few patients, however, had pulmonary mucormycosis in the present study. The diagnosis of pulmonary mucormycosis is challenging. Clinical signs are not specific, and imaging can also be non-specific. The reverse halo sign, the characteristic sign of mucormycosis, was seen only in 9.8% of the patients in a study by Feng and Sun, 2018 [89]. 

Cutaneous mucormycosis is considered to be the third most common form of mucormycosis, following the rhinocerebral and pulmonary ones. A substantial number of our patients had cutaneous mucormycosis. Many of these patients were previously healthy, and the only risk factor was trauma, usually after motor vehicle accidents. In a recent systematic review on global cutaneous mucormycosis, 15% of all cases reviewed, were secondary to motor vehicle accidents [4]. 

In the present study, 15 cases (62.5% of all cases with primary cutaneous mucormycosis) were iatrogenic. Most of them occurred in the years 2006–2007 and were associated with contaminated wound dressings. After 2009, only two cases were reported in our survey, but a series of four incidents in haematological patients was published [45], which were associated with adhesive polyethylene tapes used to stabilise peripheral venous catheters. The above-mentioned systematic review [4] has also shown that 15.6% of the reported cases in the literature were healthcare associated, some of them occurring as nosocomial outbreaks [90,91]. This is an alarming number and could be reduced, provided that a high index of suspicion exists if necrotic lesions appear in proximity to a postoperative wound, and proper measures are taken, such as direct tissue examination and tissue culture and histopathology. It should still be considered to use commercially sterile tapes or institutional tape sterilisation in haematological and neonatal units, as suggested [45,55]. Moreover, non-sterile material sites should be kept exceptionally clean and surveillance cultures from storage surfaces in units with immunosuppressed or critically ill patients should be performed periodically [45]. 

Two further cases in our survey occurred after natural disasters. One patient was trapped in her flooded house until rescued, having remained four hours under muddy water while sustained injuries on both legs. She was treated for and saved from near drowning. Her wounds were cleaned and sutured, but 72 h after admission, mucormycosis developed in all sutured wounds. Despite multiple debridements and a high dose of liposomal AmB, the infection expanded rapidly, and she died on the 6th day of hospitalisation. Another patient had extended burns after a wildfire, also in her lower extremities. After multiple debridements and a high dose of liposomal AmB for 30 days, her condition improved without requiring amputation. The cause of her infection was *L. corymbifera*. Infections in survivors of natural disasters are multimicrobial, mostly bacterial. Given the similarity to bacterial infections, particularly during the early stages of infection, mucormycosis can be misdiagnosed. With climate change and global warming contributing to greater frequency and severity of extreme weather events [92,93], the numbers of disaster-associated fungal infections, mucormycosis included, may increase. Therefore, efforts should be made to increase the index of suspicion. Direct tissue examination, tissue culture, and histopathology are crucial for early diagnosis and treatment.

In the present survey, cultures were positive in 85.3% of cases, a rate much higher than previously reported (about 50%) [7]. This was probably due to the prospective nature of the study, where in most cases, instructions were given for proper handling of the fresh tissues even before surgery. Furthermore, homogenisation before culturing was avoided. The most frequent mucoralean genus in our cohort was *Rhizopus*, usually isolated from rhinocerebral infections and was associated with the highest mortality (60.5%), with *R. arrhizus* causing most infections. *Lichtheimia* followed, although much less frequently, and resulted in a mortality of 42.9%. Of note, in older literature reports, most of the unidentified mucormycetes were designated as “*Mucor*”, giving the impression that the frequency of this genus was high. In this study, four isolates were molecularly identified as *M. circinelloides* and an additional strain had the morphologic characteristics of *Mucor*. All three patients with known outcome infected with this genus survived. Two of them had cutaneous infections, and one had sinusitis, with nasal polyps as the only predisposing factor. The rare genera *Saksenaea*, *Apophysomyces*, and *Syncephalastrum* were isolated mostly from skin/soft tissue infections and caused less mortality. There were fewer unidentified Mucorales during the second period, due to increased availability of identification methods.

The therapeutic approach to mucormycosis is multimodal, with an equally important three-point strategy, including antifungal therapy, surgical debridement, and correction of the underlying condition predisposing the patient to the disease. Antifungal therapy options are limited due to the intrinsic resistance of Mucorales to several antifungals, including voriconazole and the echinocandins [7]. In in vitro studies using standardised methods for antifungal susceptibility testing (AST), AmB is the most potent antifungal, followed by posaconazole and isavuconazole [7,94,95,96]. The use of AST to guide antifungal treatment, however, is only moderately recommended [7]. This recommendation is based on the fact that no interpretive clinical breakpoints have been defined so far, due to the difficulty in demonstrating a correlation between in vitro susceptibility testing and clinical response. AST may be clinically useful in cases of treatment failure, and it is strongly recommended primarily to establish epidemiological knowledge in the field [7]. As the first-line antifungal drug of choice in mucormycosis is liposomal AmB [7], AST was not routinely performed in the present study. In a previous study, however, based on 21 mucormycetes isolated from our patients, we showed that AmB was the most potent drug, with MICs lower than 0.5 mg/L for most of the strains tested, followed by posaconazole [97]. These results were in line with those observed in other studies using isolates of mucormycetes collected worldwide [94,95,96]. 

More than 85% of our patients were treated according to previously existing literature and updated recommendations [7,98]. Most of the cases received liposomal AmB, with or without oral posaconazole, and surgery took place whenever feasible. Surgery, in order to be effective, has to be performed as early as possible, presupposing early diagnosis, when the extension of the infection is small and the tissue damage minimal. The outcome of our cases depended highly on the extension and the site of infection. Except for patients with disseminated mucormycosis for whom mortality was 100%, and those with drug-refractory haematologic malignancy who succumbed eventually to their underlying disease, patients with the rhinocerebral form diagnosed early, survived. This was more prominent during the second part of the study in which mortality decreased significantly, from 62.8% during 2005–2008 to 34.9% after 2009. All-cause mortality rates for mucormycosis in the literature range from 40% to 80%, depending on the underlying condition and site of infection [7]. The short time from diagnosis to death in our cohort shows the remarkably rapid progression of this infection.

A limitation in the present study is that the incidence of mucormycosis in Greece remains only an estimate. As mucormycosis is not a notifiable disease and reporting was voluntary, there may have been missed cases, or cases that remained undiagnosed, such as lung infections in terminally ill haematological or SO cancer patients, in whom no post-mortem examinations were performed. These, however, are estimated to be very few, as found in a 20-year autopsy study in patients with haematologic malignancies [99]. Another limitation of the study is that several data and final outcomes are missing in a number of cases. As our main contacts for obtaining data were the Microbiology departments of the hospitals, we did not always have full access to the patients’ records. We believe, however, that those available to us are representative enough so as not to alter the results.

## 5. Conclusions

Our study gave an overview of the development and current status of mucormycosis in Greece. We showed that the incidence has remained low and stable for the last 14 years, mostly affecting patients with haematologic malignancy and neutropenia, but also a large number of individuals with type 2 diabetes, which should alert clinicians for timely diagnosis of mucormycosis in this patient population. Rhinocerebral infection was the most frequent clinical manifestation, which was less extended in the more recent period of the study and resulted in better outcomes, probably due to increased awareness and timely and proper management. Cutaneous mucormycosis was the second most frequent manifestation in our survey, affecting mostly immunocompetent patients after serious trauma, or were iatrogenic. Healthcare-associated cutaneous mucormycosis is in a major part preventable, provided that proper and high-quality wound-dressing material is used and proper measures for wound care are implemented. Increased awareness of mucormycosis is crucial, therefore monitoring of this group of infections should be continuous. 

## Figures and Tables

**Figure 1 jof-09-00425-f001:**
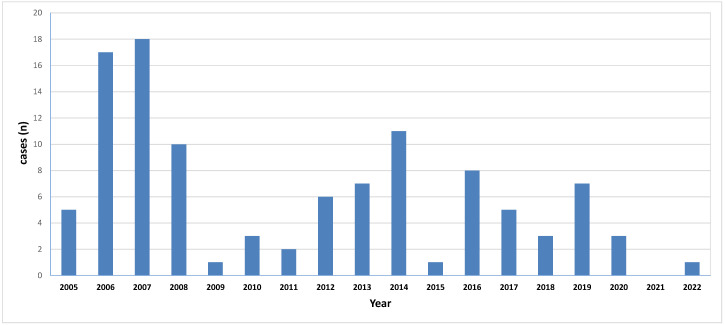
Annual incidence of mucormycosis in Greece based on the prospective survey.

**Figure 2 jof-09-00425-f002:**
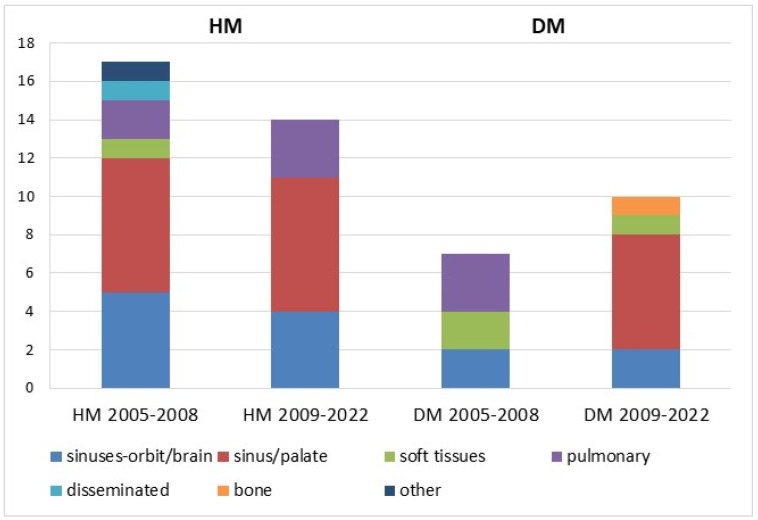
Localisation of mucormycosis in patients with haematologic malignancy (HM) or diabetes mellitus (DM) before and after 2009.

**Figure 3 jof-09-00425-f003:**
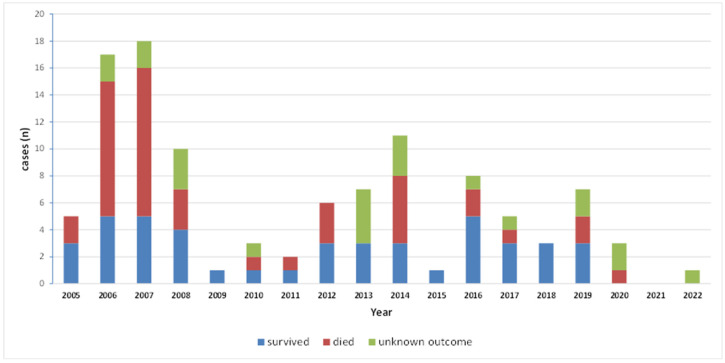
Outcome of mucormycosis cases per year.

**Table 1 jof-09-00425-t001:** Mucormycosis cases (*n* = 74) published in Greek and foreign Journals (1984–2022) and not included in the prospective survey.

Reference	Age	Gender	Localization/Clinical Form	Organism	Underlying Condition	Time to Antifungal (days)	Therapy (mg/kg/d)	Surgery	Outcome 60 d
1984 Rassias et al. [20]	x	x	Rhinocerebral	x	DM				x
1985 Filippidis et al. [21]	24	Μ	Disseminated		x				d (post mortem diagnosis)
1987 Kalafrenzos et al. [22]		x	Soft tissues	*Absidia*	none		AmB	yes	s
1988 Kotzamanoglou et al. [23]	76	F	Orbital	x	DM after cortico	10	AmB	no	d
1995 Economopoulou et al. [24]	55	F	Sinus	x	DM, cortico, pericarditis	21	AmB	no	s
1998 Athanasiou et al. [25]		x	Soft tissues	x	ALL		AmB		s
1999 Papadogeorgakis et al. [26]	65	Μ	Rhino- orbital	x	DM, COPD	60	AmB	yes	s
1999 Kanellopoulou et al. [27]	60	F	Cutaneous abdomen LE	*Rhizopus arrhizus*	none	1	AmB3	yes	s
2000 Alexopoulou et al. [28]	60	M	Rhino- orbital	*Rhizopus arrhizus*	ALL, DM	1	AmB	yes	s
2000 Tsaousis et al. [29]	73	F	Liver-Brain	Rhizopus	DM	20	AmB1.5, itr	no	s
2001 Papadaki. et al. [30]	73	F	Orbital	x	immunosuppression	>14	AmB, pos	yes	d
2001 Tsoutsos et al. [31]	38	M	Cutaneous	x	Burns	x	AmB	yes	d
2002 Tryfon et al. [32]	57	F	Rhinocerebral	x	DM, periodontitis	4	AmB0.25	no	d (10 days)
2002 Kyrmizakis et al. [33]	79	M	Palate	*Absidia corymbifera*	DM, chronic kidney failure		AmB	yes	s
2002 Akritidis et al. [34]	75	F	Cutaneous	*Rhizopus arrhizus*	DM, MDS, neutropenia	4	AmB	yes	d (14 days)
2003 Georgopoulou et al. [35]	58	F	Rhinocerebral	x	cirrhosis, renal failure, anaemia, thrombocytopenia, DM	4	AmB5	no	d (5 days)
2003 Kofteridis et al. [36]	66, 79	2M	Rhino-orbito-cerebral	x	1DM, ARF, 1DM, cortico	3, 7	AmB3	yes	s 2
2003 Petrikkos et al. [5]	37–80	15M, 9F	12 Soft tissues, 11 Rhinocerebral, 1 lung	8 *Rhizopus* spp., 16 Mucor spp.*	7DM, 1 kidney transplantation, 3 car accidents, 1 burns, 2 scratch, 1thorn (skin), 9 (F) haem (6 AML, 1CLL, 2MDS)		24 AmB3-5	11	s 11, d 13
2006 Economopoulos et al. [37]	33	F	Pulmonary.	x	DM ketoacidosis	1	AmB, +pos	yes	s
2006 Kotoulas et al. [38]	33	F	Pulmonary	x	DM coma	1	AmB, +pos	yes	lost to followup
2006 Liapis et al. [39]	38	M	Iliac artery stent	x	previous kidney graft rejection, hemodialysis, stents rec thrombosis	60	AmB3-5	yes	s
2007 Kontogiorgi et al. [40]	25	M	Pulmonary- Cutaneous	*Rhizopus* spp.	farmer tractor accident	3	AmB4-7	yes	d
2008 Mourkas et al. [41]	36	Μ	Sinus-palate	*x*	Hodgkin’s lymphoma	7	AmB, pos, hyperb	yes	s
2008 Bibashi et al. [42]	10	M	Pulmonary	*Cunninghamella*	ALL, neutropenia	1	AmB7		d
2010 Papadogeorgakis et al. [43]	22	F	Sinus	x	DM	>7	AmB, pos		s
2011 Kompoti et al. [44]	30	Μ	Cutaneous	*Saksenaea*	car accident	0	AmB, pos, hyperb	yes	d
2012 Lalayanni et al. [45]	15–58	2M, 2F	4 outbreak cutaneous	*Rhizopus arrhizus*	4 haematol		1AmB3-5, 1pos, AmB + pos2	yes	s
2012 Bibashi et al. [46]	32	M	Subcutaneous	*Lichtheimia ramosa*	car accident		no	yes	s
2014 Dimaka et al. [47]	87	M	Sinus	*x*	DM, coronary, CVA		AmB 5, pos	no	s
2015 Kyriopoulos et al. [48]	26–71	6M	Cutaneous	3 *Rhizopus, 3 Rhizomucor*	3 fire burns, 1 friction burn car accident, DM, 2 road accidents	0–21	AmB0.5	5yes	d 2 fire burns
2018 Kalogerakos et al. [49]	56	M	Thoracoabdominal	*x*	AML, neutropenia		AmB + pos	yes	s
2019 Samaras et al. [50]	39	M	Cutaneous	*Saksenaea*	injury	63	AmB 5, pos		s
2019 Dotis et al. [51]	16	F	Disseminated	*Rhizopus arrhizus*	kidney transplant		AmB 5, pos	yes	s
2019 Athanasiadou et al. [52]	11	M	Rhino-orbital	*Rhizopus arrhizus*	recurrent neuroblast, neutrop.	4	AmB 7, +pos		s
2020 Antoniadi et al. [53]	2–15	2F 4M	3 Rhino-orbital, 2 CNS, 1 Soft tissues	*2 Rhizopus, 1 Mucor* sp., *1 Lichtheimia* sp.	4 paediatric malignancy (3 ALL, 1 AML,1 neuroblastoma, 1 CNS tumor	5 to 30 (avg. 13)	1L-AmB3-10, 1 + pos, 2 + cas, 1 + vor, 1 + cas, vor	4yes	s 6
2021 Stavrakas et al. [54]	52	M	Sinus	*x*	HIV	1	yes	yes	s
2022 Ziaka et al. [55]	10	F	Cutaneous	*Rhizopus microsporus*	ALL	1	AmB8	yes	s

x; not reported, d; died, s; survived, M; male, F; female, AML; acute myeloid leukaemia, ALL; acute lymphocytic leukaemia, CLL; chronic lymphocytic leukaemia, MDS; myelodysplastic syndrome, DM; diabetes mellitus, ARF; acute renal failure, COPD; chronic obstructive pulmonary disease, CNS; central nervous system, CVA; cerebrovascular accident, AmB; amphotericin B, pos; posaconazole, cas; caspofungin, itr; itraconazole, vor; voriconazole.

**Table 2 jof-09-00425-t002:** Clinical forms of mucormycosis in registered patients.

	All Cases		2005–2008		2009–2022	
Clinical Form/Localization	n	%	n	%	n	%
Rhinocerebral	56	51.8	22	44	34	58.6
Cutaneous/soft tissues	35	32.4	20	40	15	25.9
Pulmonary	12	11.1	6	12	6	10.3
other *	2	1.8	1	2	1	1.7
Disseminated	2	1.8	1	2	1	1.7
Bone **	1	0.9		2	1	1.7
Total	108		50		58	

* brain, lung-heart. ** mastoid.

**Table 3 jof-09-00425-t003:** Underlying conditions/risk factors in registered patients with mucormycosis.

Underlying Condition	All Cases		2005–2008		2009–2022	
	n	%	n	%	n	%
Haematologic malignancy/neutropenia	32	**29.9**	17	**34**	15	**26.3**
AML	13		5		8	
non-Hodgkin’s lymphoma	6		4		2	
MDS	4		3		1	
Hodgkin lymphoma	2		1		1	
aplastic anemia	2		2			
CMML	2				2	
ALL	1		1			
CLL	1				1	
ABL	1		1			
HSCT	5	**4.7**	1	**2**	4	**7**
Kidney transplantation	3	**2.8**			3	**5.3**
Diabetes mellitus	17	**15.9**	7	**14**	10	**17.5**
Trauma	24	**22.4**	15	**30**	9	**15.8**
Burns	2				2	
Surgery/iatrogenic	15		13		2	
Accidents	7		2		5	
SO/or other cancer	9	**8.4**	3	**6**	6	**10.5**
Auto-immune	8	**7.5**	6	**12**	2	**3.5**
other *	5	**4.7**	1	**2**	4	**7**
None/not reported	4	**3.7**			4	**7**
Total **	107		50		57	

* chronic renal failure/haemodialysis, thalassaemia, pulmonary infection/cardiopulmonary ICU, nasal polyps. ** one patient had 2 infections, both due to the same underying condition (nasal polyps). AML; acute myeloid leukaemia, MDS; myelodysplastic syndrome, CML; chronic myelomonocytic leukaemia. ALL; acute lymphocytic leukaemia, CLL; chronic lymphocytic leukaemia, ABL; acute biphenotypic leukaemia. HSCT; Haematopoietic stem cell transplantation, SO; solid organ.

**Table 4 jof-09-00425-t004:** Clinical forms of mucormycosis in patients with haematologic malignancy (HM) and diabetes mellitus (DM).

Clinical Form	HM n = 32	%	DM n = 17	%
Rhinocerebral	23	71.9	10	58.8
Cutaneous			3	17.6
Pulmonary	5	15.6	3	17.6
Disseminated	2	6.2		
Other *	2	6.2	1	5.9

* lung/heart, submandibular area.

**Table 5 jof-09-00425-t005:** Mucoralean genera/species isolated from registered mucormycosis cases and all-cause mortality according to genus (deaths/n).

Organisms	Cases		ROC	Pulmonary	Cutaneous	Other	Mortality (%)
	n	%					All
*Rhizopus*	**55**	**67.1**	14/36	3/4	7/14	1/1 disseminated	26/43 * (60.5)
*R. arrhizus*	45	54.9					
*R. microsporus*	3	3.6					
*Rizopus* spp.	7	8.5					
*Lichtheimia*	**7**	**8.5**	0/1	1/1	1/3	0/1 brain, 1/1 lung/heart	3/7 (42.9)
*L. ramosa*	2	2.4					
*L. corymbifera*	1	1.2					
*Lichtheimia* spp.	4	4.9					
*Mucor*	**5**	**6.1**	0/1		0/2		0/3 * (0)
*M. circinelloides*	4	4.9					
*Mucor* spp.	1	1.2					
*Saksenaea vasiformis*	**3**	**3.6**			0/2	0/1 bone	0/3 (0)
*Apophysomyses elegans*	**1**	**1.2**			0/1		0/1 (0)
*Syncephalastrum racemosum*	**1**	**1.2**			0/1		0/1 (0)
Mucorales **	**10**	**12.2**					
**Total**	**82**						

ROC; rhinocerebral. * cases with known outcome, infected with the given organism. ** unidentified genus.

**Table 6 jof-09-00425-t006:** Outcome according to localisation/clinical form and underlying condition.

		2005–2008					2009–2022			
Localisation	Mortality	%	Survival	%	Unknown	Mortality	%	Survival	%	Unknown
Sinus/palate	2/10	28.6 *	5/10	71.4 *	3	1/18	7.1 *	13/18	92.8 *	4
Rhino-(orbito)-cerebral	10/12	90.9 *	1/10	9.1 *	1	8/15	72.7 *	2/15	27.3 *	5
Cutaneous	9/20	52.9 *	8/20	47.1 *	3	3/15	25 *	9/15	75 *	3
Pulmonary	4/6	66.7	2/6	33.3		1/6	33.3 *	2/6	66.7 *	3
Disseminated	1/1	100				1/1	100			
Lung/heart	1/1									
Brain						1/1	100			
Bone								1/1	100	
**Underlying condition**										
HM/neutropenia	10/18	66.7 *	5/18	33.3 *	3	4/15	40 *	6/10	60 *	5
HSCT					1			2/4	50 *	2
Diabetes mellitus	2/7	33.3 *	4/7	66.7 *	1	2/10	20	8/10	80	
Trauma	8/15	57.1 *	6/15	42.9 *	1	3/9	33.3	6/9	66.7	
SO/or other cancer	3/3	100				3/6	50	3/6	50	
Autoimmune	4/6	80 *	1/6	20 *	1	1/2	100 *			1
Kidney transplantation						1/3	100 *			2
Other **					1	1/8	33.3 *	2/8	66.7	5

"HM; haematologic malignancy, HSCT; haematopoietic stem cell transplantation, SO; solid organ." * calculated on cases with known outcome. ** thalassaemia, chronic renal failure/haemodialysis, pulmonary infection/cardiopulmonary ICU, nasal polyps, none, unknown.

**Table 7 jof-09-00425-t007:** Differences between published (2005–2022) and registered cases.

	Published	%	Registered	%
**Localisation**				
Rhinocerebral	8	25	56	51.5
Cutaneous/soft tissues	17	53.1	35	32
Pulmonary	4	12.5	12	10.3
CNS	2	6.2	1	2
Disseminated	1	3	2	3.1
other ^a^			2	3.1
**Underlying condition**				
Haematologic malignancy	12	37.5	32	29.9
Kidney transplantation	2	6.2	3	2.8
Diabetes mellitus	4	12.5	17	15.9
Trauma/burn	10	31.2	24	22.4
SO cancer	3	9.4	9	8.4
Autoimmune			8	7.5
HIV	1	3		
**Organism**				
*Rhizopus arrhizus*	6	25	45	54.9
*Rhizopus microsporus*			3	3.6
*Rhizopus* spp.	7	29.2	7	8.5
*Lichtheimia* spp.	2	8.3	7	8.5
*Mucor* spp.	1	4.2	5	6.1
*Saksenaea vasiformis*	2	8.3	3	3.6
*Apophysomyces elegans*			1	1.2
*Syncephalastrum racemosum*			1	1.2
*Cunninghamella bertholetiae*	1	4.2		
*Rhizomucor*	3	12.5		
*Mucorales* *	2	8.3	10	13.9
**Outcome**				
survival	26	83.9 **	44	51.2 **
death	5	16.1 **	42	48.8 **
unknown	1	2.7	22	20.4

CNS; central nervous system, SO; solid organ, ^a^ lung-heart. * unidentified genus. ** calculated on cases with known outcome.

## Data Availability

Anonymised data are available by the corresponding author upon reasonable request.

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
