# Peer review of "Epidemiology of Mucormycosis in Greece; Results from a Nationwide Prospective Survey and Published Case Reports"

_jof, 2023, doi:10.3390/jof9040425_

Round 1

Reviewer 1 Report

1.  The manuscript under peer-review entitled, “Epidemiology of mucormycosis in Greece; results from a nationwide prospective survey and published case reports” describes the uncommon disease, however it is not that uncommon now, particularly after the Covid-19 pandemic. The authors have analysed the epidemiology of mucormycosis in Greece in multicenter, nation-wide prospective survey of patients of all ages during 2005-2022, wherein a total of 108 cases were recorded.

2.     The manuscript drafting is very good and elaborative. The subject matter has been introduced in a nice manner, which deals with the reports from Greek hospitals and also retrieved from the literature. The gist of cases has been mentioned in the manuscript, same is elaborated in the Discussion and corresponding reports are cited under the heading of References.

3.      The Results are well-documented in the text as well as in the tabulated form. The Table 1 narrates 74 mucormycosis cases published in Greece and foreign journals, with all their clinical descriptions, organisms isolated, underlying condition, duration of treatment given. The Figure 1 shows the annual incidence of mucormycosis in Greece based on the prospective survey.

4.      Another Table shows the clinical forms of mucormycosis in patients with haematologic malignancy and diabetes mellitus. The maximum cases were of rhino-cerebral mucormycosis followed by pulmonary, cutaneous and disseminated forms. The Figure 2 shows the comparative localisation of mucormycosis in patients with haematologic malignancy or diabetes mellitus before and after 2009.

5.   Maximum cases were caused by Rhizopus species followed by Lichtheimia Mucor, Saksenaea, Apophysomyces species. Figure 3 shows the outcome of mucormycosis cases per year like their survival or demise or in some cases with the unknown outcome.

6.         Many countries faced the upsurge of mucormycosis cases (CAM) like India during the 2nd wave of Covid-19 in May-June 2021, wherein more than 51,000 cases were reported. However, this type of upsurge was not seen in Greece.

7.      The Discussion being focused on the clinical and laboratory findings of all clinical cases mentioned in this manuscript, in the light of contemporary studies, which have been cited in the text. Most of the cited studies have similar results as that of the current study.

8.      The References are more than adequate in number and the authors have tried to justify with citations vis-a-vis the clinical findings of the relevant case and/or isolate mentioned in the manuscript as well as comparing contemporary studies by other workers on the subject matter. Almost all the reported cases of mucormycosis from Greece have been included in the list of References.

Author Response

Reviewer 1

Comments and Suggestions for Authors

  1. The manuscript under peer-review entitled, “Epidemiology of mucormycosis in Greece; results from a nationwide prospective survey and published case reports” describes the uncommon disease, however it is not that uncommon now, particularly after the Covid-19 pandemic. The authors have analysed the epidemiology of mucormycosis in Greece in multicenter, nation-wide prospective survey of patients of all ages during 2005-2022, wherein a total of 108 cases were recorded.

We thank the reviewer for taking the time to review our paper and provide useful comments. A sentence about Covid-19 pandemic has been added in the Introduction.

  1. The manuscript drafting is very good and elaborative. The subject matter has been introduced in a nice manner, which deals with the reports from Greek hospitals and also retrieved from the literature. The gist of cases has been mentioned in the manuscript, same is elaborated in the Discussion and corresponding reports are cited under the heading of References.

We thank the reviewer for the kind words and positive comments. We understand that no further revisions are required at this point.

  1. The Results are well-documented in the text as well as in the tabulated form. The Table 1 narrates 74 mucormycosis cases published in Greece and foreign journals, with all their clinical descriptions, organisms isolated, underlying condition, duration of treatment given. The Figure 1 shows the annual incidence of mucormycosis in Greece based on the prospective survey.

We thank the reviewer for the positive comments. We understand that no further revisions are required at this point.

  1. Another Table shows the clinical forms of mucormycosis in patients with haematologic malignancy and diabetes mellitus. The maximum cases were of rhino-cerebral mucormycosis followed by pulmonary, cutaneous and disseminated forms. The Figure 2 shows the comparative localisation of mucormycosis in patients with haematologic malignancy or diabetes mellitus before and after 2009.

We thank the reviewer for highlighting our results. We understand that no further revisions are required at this point.

  1. Maximum cases were caused by Rhizopus species followed by Lichtheimia Mucor, Saksenaea, Apophysomyces species. Figure 3 shows the outcome of mucormycosis cases per year like their survival or demise or in some cases with the unknown outcome.

We thank the reviewer for highlighting our results. We understand that no further revisions are required at this point.

  1. Many countries faced the upsurge of mucormycosis cases (CAM) like India during the 2nd wave of Covid-19 in May-June 2021, wherein more than 51,000 cases were reported. However, this type of upsurge was not seen in Greece.

We thank the reviewer for highlighting this finding. A sentence regarding CAM has been added in the Discussion.

  1. The Discussion being focused on the clinical and laboratory findings of all clinical cases mentioned in this manuscript, in the light of contemporary studies, which have been cited in the text. Most of the cited studies have similar results as that of the current study.

We thank the reviewer for the positive comment. We understand that no further revisions are required at this point.

  1. The References are more than adequate in number and the authors have tried to justify with citations vis-a-vis the clinical findings of the relevant case and/or isolate mentioned in the manuscript as well as comparing contemporary studies by other workers on the subject matter. Almost all the reported cases of mucormycosis from Greece have been included in the list of References.

We thank the reviewer for the kind words. We understand that there are no further points to be revised.

Reviewer 2 Report

·         Although it is of undoubted epidemiological interest, the present work appears to be scattered and not well structured. In particular, the authors describe collected data in a non-homogeneous and very verbose way, alternating each paragraph reporting the data collected in the prospective study with those reported from the literature data. My suggestion is to streamline the text, taking advantage of the tables and grouping most of the literature data in one section, preferably in the introduction.

·         In table 1, the acronyms of the last column have not been specified (i.e. d,s).

·         I do not fully agree with the authors' decision to divide the surveillance period into two parts: 2005-2008 and 2009-2022. Although it could be that since 2009 diagnostic and treatment techniques were improved, it is a risky and not very useful choice. The authors could have simply report the collected data without dividing the two periods, given that no particular statistical analyzes were carried out either. In fact, the only statistical calculation that was made by the authors is that of p on mortality (line 309 of the text)

Author Response

Reviewer 2

Comments and Suggestions for Authors

  • Although it is of undoubted epidemiological interest, the present work appears to be scattered and not well structured. In particular, the authors describe collected data in a non-homogeneous and very verbose way, alternating each paragraph reporting the data collected in the prospective study with those reported from the literature data. My suggestion is to streamline the text, taking advantage of the tables and grouping most of the literature data in one section, preferably in the introduction.

We thank the reviewer for taking the time to review our paper and provide constructive criticism. We agree that it would be better to group most of the literature data in one section. As we consider them as Results, we have done so in the Results section, under Chapter 3.3. We also tried to streamline the text as much as we could, removing detailed data that can be seen in the tables.

  • In table 1, the acronyms of the last column have not been specified (i.e. d,s).

Table 1 has been updated.

  • I do not fully agree with the authors' decision to divide the surveillance period into two parts: 2005-2008 and 2009-2022. Although it could be that since 2009 diagnostic and treatment techniques were improved, it is a risky and not very useful choice. The authors could have simply report the collected data without dividing the two periods, given that no particular statistical analyzes were carried out either. In fact, the only statistical calculation that was made by the authors is that of p on mortality (line 309 of the text).

We thank the reviewer for this comment. In fact, this issue concerned us as well, however, we considered it more appropriate to divide the data in the selected two periods for the following reasons: a) data of a small period at the beginning of the prospective study that were very different from those of the following years, would affect the end results making them less accurate and reliable, b) Since a difference in incidence was observed, we considered it imperative to analyse it further, and highlight the variables that contributed to that change, c) The only statistically significant result was indeed that of mortality, but qualitative differences, such as less extended sinus infections, fewer iatrogenic and haematological cases in the more recent period are also, to our view, too important to be ignored. By highlighting them we had the opportunity to discuss them, and show that with increased awareness, the morbidity and mortality of these infections can be reduced, whereas many cases can be also prevented. We believe that this adds value to our research, and this is the whole essence of a long-running prospective study. Collective data are presented anyway.

Furthermore, in the text, lines 313-314, it was mistakenly written that ‘no statistical analysis could be performed’. In fact, we did perform statistical analysis for all variables, but the results were not statistically significant. It should be noted however, that the sample size per category was very small.  We corrected the text accordingly.  

Reviewer 3 Report

The review deals with the epidemiology of Mucormycosis in Greece, the review is well written, anyway it needs some additional information to be added.

Major revisions

Abstract: Please add information about Mucormycosis and COVID-19 from 2020-2022

Introduction: Add a paragraph on global prevalence of COVID-19 associated mucormycosis worlwide and in Greece (if not cases are reported please cite this).

Line 125: Please add informations about methods used for standard identification at the species level (microscopic, MALDI-TOF or others) and molecular identification (primers, matching databases, etc).

What about susceptibility tests? Which percentage of samples were subjected to AST and which are the reported susceptibilities for the different antifungals, please add a paragraph and a table if appropriate with data from literature.

Discussion; Add considerations about susceptibility tests and treatments.

Author Response

Reviewer 3

The review deals with the epidemiology of Mucormycosis in Greece, the review is well written, anyway it needs some additional information to be added.

Major revisions

Abstract: Please add information about Mucormycosis and COVID-19 from 2020-2022

We thank the reviewer for taking the time to review our paper and provide useful comments. Due to word number restrictions, we were not able to add information on Mucormycosis and COVID-19 in the Abstract, but we added a sentence in the Introduction and more information in the Discussion.

Introduction: Add a paragraph on global prevalence of COVID-19 associated mucormycosis worlwide and in Greece (if not cases are reported please cite this).

We have added a sentence in the Introduction and more information with relevant reference on global prevalence of COVID-19 associated mucormycosis worldwide in the Discussion, where we state that no CAM cases have been reported in Greece so far.

Line 125: Please add informations about methods used for standard identification at the species level (microscopic, MALDI-TOF or others) and molecular identification (primers, matching databases, etc).

We thank the reviewer for this useful comment. Information on molecular identification with the relevant reference has been added.

What about susceptibility tests? Which percentage of samples were subjected to AST and which are the reported susceptibilities for the different antifungals, please add a paragraph and a table if appropriate with data from literature.

We thank the reviewer for this comment. Unfortunately, as susceptibility testing is not recommended by global guidelines, it was not routinely performed. However, we have conducted a relevant study in the past, published in J Antimicrob Chemother 2012; 67: 1937–1940. We avoided to repeat those findings in the present study, but following the reviewer’s suggestion we added a sentence in the Discussion.

Discussion; Add considerations about susceptibility tests and treatments.

We added those considerations in the Discussion, as suggested.

Reviewer 4 Report

This is a nice analysis of the epidemiology of mucormycosis is Greece. 

Some comments and modifications to consider 

Line 53 - about DM does not make sense. Please review to make it clear.  I think you are trying to say that the DM was due to the underlying condition or the use of steroids but not 100% sure. Please make it crystal clear.

Line 57 - with or without posaconazole. There may not be room in the abstract and so may need to do this in the main text. But it would be good to know the percentage that got combination therapy with posaconazole and the percentage that got sequential therapy with posaconazole  (after a course of LAMB).

Line 87 - voluntary registration. This may result in under-reporting. Need to say that this could be a limitation in the discussion. You do talk about under-reporting but don't specifically refer to the voluntary nature in the discussion. Needs to be made more explicit.

Line 91 - should be case report form (no capitals needed)

Line 97 - take out either herein or presented. Otherwise it doubles up

Line 99-100 - need to make this clear that the investigators got instructions in how to manage the culture - optimal processing 

Line 114 - it would be good to say here the number that were retrieved

Line 201 - same comment as above about DM and comorbidity

287-289 - needs more information. Need duration L-AmB and percentage that had combination posaconazole and percentage that had posaconazole sequentially

Line 344- an extra space that need to be eliminated

Discussion  - break up the paragraphs more. This will make it easier to read and digest.

Conclusion - needs shortening. Too long for a clonclusion.        

Author Response

Reviewer 4

This is a nice analysis of the epidemiology of mucormycosis is Greece. 

Some comments and modifications to consider 

Line 53 - about DM does not make sense. Please review to make it clear.  I think you are trying to say that the DM was due to the underlying condition or the use of steroids but not 100% sure. Please make it crystal clear.

We thank the reviewer for their kind words. We agree that the way this sentence was written might not be clear enough. In fact, these patients had DM at the same time with the main underlying condition, and not due to the underlying condition. In order to avoid confusion, the words ‘of the main condition’ have been deleted and the sentence has been rephrased.

Line 57 - with or without posaconazole. There may not be room in the abstract and so may need to do this in the main text. But it would be good to know the percentage that got combination therapy with posaconazole and the percentage that got sequential therapy with posaconazole  (after a course of LAMB).

We thank the reviewer for highlighting this important point. We included these data in the Results.

Line 87 - voluntary registration. This may result in under-reporting. Need to say that this could be a limitation in the discussion. You do talk about under-reporting but don't specifically refer to the voluntary nature in the discussion. Needs to be made more explicit.

The word ‘voluntary’ in the Discussion has been added.

Line 91 - should be case report form (no capitals needed)

Corrected.

Line 97 - take out either herein or presented. Otherwise it doubles up.

Corrected.

Line 99-100 - need to make this clear that the investigators got instructions in how to manage the culture - optimal processing.

The change has been made. 

Line 114 - it would be good to say here the number that were retrieved

The number is presented as a result, in the Results section.

Line 201 - same comment as above about DM and comorbidity

It has been clarified, as above.

287-289 - needs more information. Need duration L-AmB and percentage that had combination posaconazole and percentage that had posaconazole sequentially

The information has been added.

Line 344- an extra space that need to be eliminated.

Corrected.

Discussion  - break up the paragraphs more. This will make it easier to read and digest.

We tried to follow this suggestion as much as possible.

Conclusion - needs shortening. Too long for a clonclusion. 

The conclusion section was shortened. 

Round 2

Reviewer 2 Report

I agree with the authors' changes that improve the quality of the work.

Author Response

We thank the reviewer for agreeing with our changes. 

Reviewer 3 Report

The authors declared that:

as antifungal susceptibility testing (AST) to guide treatment is only marginally recommended [7], we did not perform AST routinely.

Anyway this is what is reported from the guidelines:

The use of standard methods for antifungal susceptibility testing to guide antifungal treatment in Mucorales is marginally supported and may be clinically useful in cases of treatment failure. However, we strongly recommend the use of these methods primarily to establish epidemiological knowledge in the field. 

The authors should collect data about susceptibility worldwide, beside their experience, and cyte all the data from literature. They have to change the sentence about guidellines following what is reported for clinical failure and the strong recommendation for susceptibility tests.

Author Response

Review report (Round 2)

Reply to Reviewer 3

The authors declared that:

as antifungal susceptibility testing (AST) to guide treatment is only marginally recommended [7], we did not perform AST routinely.

Anyway this is what is reported from the guidelines:

The use of standard methods for antifungal susceptibility testing to guide antifungal treatment in Mucorales is marginally supported and may be clinically useful in cases of treatment failure. However, we strongly recommend the use of these methods primarily to establish epidemiological knowledge in the field. 

-We thank the reviewer for the suggestion, we added those recommendations to the text in the Discussion.

The authors should collect data about susceptibility worldwide, beside their experience, and cyte all the data from literature. They have to change the sentence about guidellines following what is reported for clinical failure and the strong recommendation for susceptibility tests.

-We modified our text in the Discussion according to the reviewer’s suggestions, with the relevant citations. As AST was not part of the present study, we tried to keep our text as short as possible.